# Anti-Cancer and Anti-Inflammatory Properties of Black Garlic

**DOI:** 10.3390/ijms25031801

**Published:** 2024-02-01

**Authors:** Agnieszka Ewa Stępień, Julia Trojniak, Jacek Tabarkiewicz

**Affiliations:** 1Institute of Health Sciences, Medical College of Rzeszów University, University of Rzeszow, 35-959 Rzeszów, Poland; astepien@ur.edu.pl; 2Student’s Scientific Club of Immunology, Institute of Medical Sciences, Medical College of Rzeszów University, University of Rzeszow, 35-959 Rzeszów, Poland; jt117576@stud.ur.edu.pl; 3Department of Human Immunology, Institute of Medical Sciences, Medical College of Rzeszów University, University of Rzeszow, 35-959 Rzeszów, Poland

**Keywords:** black garlic, anti-inflammatory, anti-cancer

## Abstract

Black garlic (BG) is a fermented form of garlic (*Allium sativum L*.), produced at precisely defined temperatures, humidities, and time periods. Although garlic has been used for thousands of years, black garlic is a relatively new discovery. There are many bioactive compounds in black garlic that give it medicinal properties, including anti-inflammatory and anti-cancer properties. In our review article, we present scientific studies examining the anti-inflammatory and anti-cancer effects of black garlic. According to research, this effect is mainly due to the reduction in the production of pro-inflammatory cytokines, as well as the ability to scavenge free oxygen radicals and induce apoptosis. In addition, the phytochemicals contained in it have antiproliferative and antiangiogenic properties and inhibit the growth of cancer cells. Black garlic is a valuable source of biologically active substances that can support anti-inflammatory and anti-cancer therapy. Compared to *Allium sativum,* black garlic has fewer side effects and is easier to consume.

## 1. Introduction

Garlic (*Allium sativum L*.) is a shallow-rooted vegetable plant belonging to the *Alliaceae* family [1]. Native to western Asia and the Mediterranean coast, this plant is widely distributed around the world [2]. *Allium sativum* includes two subspecies: *A. sativum variety sativum* (softneck garlic) and *A. sativum variety ophioscorodon* (hardneck garlic) [1]. There are differences between both subspecies in terms of structure. The head of hard garlic has a hard neck and six to eleven cloves surrounding a woody stem, while soft garlic has no flower top, contains up to twenty-four cloves, and has a stem that is soft and central [1]. Garlic can grow in temperate and warm climate zones and is perennial [1,3].

Garlic has been used in traditional medicine around the world since ancient times due to its valuable health-promoting properties [4]. Scientific research results indicate a number of health-promoting properties: hepatoprotective, nephroprotective, immunomodulatory, anti-allergic, antioxidant and anti-cancer properties (Figure 1) [5,6,7,8,9]. Its unfavorable taste and smell have recently significantly reduced consumption all over the world, with the exception of China and India [10]. Consuming raw garlic in quantities to achieve enormous health benefits for the patient is difficult due to its pungent taste and odor [11].

Garlic consumption is accompanied by an unpleasant odor from the mouth and body. There is evidence that garlic consumed in large doses has toxic effects [8]. Additionally, a significant number of consumers have stopped consuming garlic due to gastrointestinal discomfort, including damage to the stomach and intestinal walls [10,11]. When large doses of garlic are used, negative side effects are observed: anemia, calcium deficiency and contact allergies [8].

In recent years, there has been an increase in interest in the health-promoting properties of black garlic (BG) as a rich source of several bioactive compounds, mainly those with antioxidant properties [4,12]. It is obtained from raw garlic subjected to aging processes [13,14]. Black garlic is produced by fermentation and ripening under strictly defined conditions of a temperature of 60–90 °C and humidity of 60–90% for 10 to 80 days [14]. As it ages under precisely defined processing parameters, e.g., temperature or humidity, the color of garlic changes from white to dark brown/black [15]. The color change is the result of enzymatic browning in the Millard reaction—condensation between the reducing carbonyl group of sugar and the amino group [10,15,16,17]. In Figure 2, we can visualize the difference between fresh garlic and aged garlic. During the fermentation process, white garlic loses its sharp taste due to the alliin content in favor of a sweet or sweet–sour taste and becomes odorless and has a consistency ranging from rubbery and stringy to gelatinous [4,14,18]. Moreover, new S-allicin compounds resulting from the transformation of allicin have very strong antioxidant properties as a result of the Maillard reaction between reducing sugars and allicin [19]. During garlic aging, the chemical oxidation of phenols and thermal degradation of organic sulfur compounds also occur [4,20]. 

## 2. Active Compounds of Black Garlic

The aging process of black garlic causes a change in physicochemical properties associated with an increase in the content of antioxidant compounds [21]. Compared to fresh garlic, black garlic has a significantly different chemical composition—as shown in Table 1. Black garlic is characterized by a high concentration of many antioxidant compounds, including the following: phenols, flavonoids, pyruvate, S-Allyl-Cysteine (SAC), S-allyl-Mercapto-Cysteine (SAMC), and 5-hydroxymethylfurfural (5-HMF). It also contained allicin-derived organosulfur compounds (OSC): diallyl sulfides (DAS), diallyl disulfides (DADS), diallyl trisulfides (DATS), and diallyl tetrasulfide [10,22]—as shown in Figure 3a–h. Scientific research results have shown that the content of phenols differs significantly between black and fresh garlic [23]. An increase in temperature and a decrease in humidity during aging significantly increases the level of polyphenols, thiosulfonates, and allicin [24].

Scientific research highlights the anti-cancer, anti-proliferative, anti-inflammatory, immunomodulatory, cardioprotective, nephroprotective and hepatoprotective properties of black garlic, its protective effects on the digestive system and nervous system, and its anti-diabetic and anti-obesity properties [35].

## 3. Antioxidative Properties of Black Garlic (BG)

Many oxidation processes occurring in living organisms generate free radicals, including reactive oxygen species (ROS). The body’s defense mechanisms neutralize them by blocking or delaying the oxidation process. It is very important to prevent the accumulation of non-neutralized free radicals because they induce oxidative stress, which increases the risk of inflammation and carcinogenesis [36].

Scientists still face challenges in developing exceptional-quality black garlic that is high in health-promoting compounds [37]. A number of bioactive substances in black garlic, including phenols and flavonoids, increase the scavenging of free radicals in aging. The antioxidant capacity of black garlic increases with processing temperature [24]. Furthermore, black garlic’s antioxidant capacity is also affected by its type, moisture, and fermentation time—as shown in Table 2. To maximize antioxidant abilities, garlic fermentation should last for 8–12 days at 70–75 °C with a moisture content of 80–85% [24,38]. Using a temperature that is lower than those mentioned will prevent the garlic from turning black and keep its pungent aroma, whereas using a temperature that is higher than those mentioned will cause the garlic to become bitter [13].

Scientific research has shown that mature black garlic extract (ABGE) has very high antioxidant properties [22]. It has also been shown that the use of mature black garlic (ABG) compared to fresh raw garlic (FRG) significantly reduces the activation of superoxide dismutase (SOD) in vitro with a high number of total phenols in the extract [10,43]. Gavilán et al. analyzed the antioxidant capacity of black and fresh garlic using DPPH and ABTS tests. They showed that, even at lower concentrations (4 mg/mL), black garlic showed greater antioxidant activity than fresh garlic [44]. The inhibition of the DPPH-free radical using 10 mg/mL of ABGE elicited results of 51 ± 5.7%, while white garlic showed only 12 ± 2.6% inhibition at the same concentration.

Research indicates differences in the content of black garlic’s chemical compounds and antioxidant capacity depending on the variety and preparation of garlic. According to some studies, black garlic from the elephant garlic variety *(Allium ampeloprasum)* shows increased antioxidant activity but contains fewer polyphenols than common garlic *(Allium sativum)* [45,46]. Further research is needed to clarify this theory, but it suggests that the antioxidant properties of BG compounds may be the result of sulfur compounds, rather than polyphenols [44]. 

There was a significant difference between water extracts and alcohol extracts of black garlic in terms of their antioxidant properties. Black garlic water extracts showed much higher antioxidant activity than alcohol extracts [47]. This may have been due to the reduction in the polarity of allicin derivatives caused by the Millard reaction, making the substances present in black garlic prefer water over other solvents [39]. 

Studies conducted on the LPS-adapted macrophage line (RAW264.7) showed that pyruvate present in black garlic inhibits the formation of ROS induced by H_2_O_2_ [48].

Less inhibition of hydroxyl radicals was observed by Farhart Z. et al. with the use of hydroxyl scavenging, DPPH, and superoxide radical scavenging assays, and antioxidant activity ranged from 0.25 to 2.01 mg AAE per g of garlic for aqueous extracts and 0 to 3.26 mg AAE per g of garlic for methanol extracts [47]. A study conducted by Kim et al. showed that black garlic extract with a concentration of (2 mg mL^−1^) showed higher antioxidant properties compared to fresh garlic extract. At a lower concentration (0.2 mg mL^−1^), the antioxidant properties of black garlic were not as strong [8]. The authors determined the scavenging activities on the basis of 1,1-diphenyl-2-picrylhydrazyl and hydroxyl radicals, ferricyanide reducing power, ferrous ion-chelating ability, and inhibitory effect on linoleic acid peroxidation.

The antioxidant capacity of mature black garlic was determined to be 4.5 times higher than that of fresh garlic using the TEAC technique. This indicated that the garlic fermentation process significantly increased its antioxidant properties [49]. TEAC values of garlic and ABG were 13.3 ± 0.5 and 59.2 ± 0.8 µmol/g wet weight, respectively

## 4. Anti-Inflammatory Properties of Black Garlic

It has been shown that black garlic significantly reduces blood sugar levels, lipid peroxidation, and antioxidant defenses by activating downstream nuclear factor erythroid 2-related factor 2 (Nrf2) and Nrf2 targets such as quinone-oxidoreductase-1 (NQO1), heme oxygenase-1 (HO-1), and glutathione S-transferase alpha 2 (GSTA2) [49].

There are several compounds in black garlic that could have anti-inflammatory effects, including pyruvate, S-Allyl-Cysteine (SAC), 2-linoleoylglycerol, and 5-hydroxymethylfurfural (5-HMF) [10,48,50]. 

It was shown that 5-hydroxymethylfurfural suppressed cell adhesion by human umbilical vein endothelial cells (HUVEC) by inhibiting the expression of vascular cell adhesion molecule-1 (VCAM-1) and intercellular adhesion molecule (ICAM-1), ROS generation, and nuclear factor kappa B activation [51]. Kong et al. investigated 5-Hydroxymethylfurfural’s (5-HMF) effects on LPS-stimulated RAW 264.7 macrophage inflammatory responses. It was found that 5-HMF suppressed the phosphorylation of proteins connected to MAPK, NF-κB, and Akt/mTOR signaling pathways. The inhibition of these pathways was mediated through inhibition of pro-inflammatory mediators (NO, PGE2, TNF-α, IL-6 and IL-1β) and reactive oxygen species (ROS) [52]. 

Using HaCaT keratinocytes as a model system, it was found that S-Allyl-Cysteine (SAC) from black garlic (BG) induces an anti-inflammatory response by inhibiting the production of pro-inflammatory cytokines TNF-α and IL-1β. In addition, S-Allyl-Cysteine significantly inhibited TNF-α-induced activation of P38 and JNK MAP kinases and NF-κB [53].

A mouse model of contact dermatitis showed that black garlic decreased the activation of macrophages, as well as the release of inflammatory mediators like nitric oxide II (NO), tumor necrosis factor (TNF-α), and interleukin 6 (IL-6) [54]. A reduction in inflammatory mediators was achieved by inhibiting iNOS, COX-2, and NF-κB. Additionally, the fraction of black garlic extract (BG10) showed a stronger anti-inflammatory effect against 12-O-tetradecanoylphorbol-13-acetate (TPA)-induced contact dermatitis in RAW264.7 cells compared to crude black garlic extract (ABG) [54]. In a study involving RAW 264.7 macrophages that had been stimulated with LPS, Kim et al. investigated the anti-inflammatory properties of BG. In aged black garlic, one compound (AGE-1) inhibited the production of pro-inflammatory mediators (NO, PGE2, IL-1β, IL-6 and TNF-α). In contrast, the second compound (AGE-2) did not show such an effect [48]. 

The anti-inflammatory and hepatoprotective effects of black garlic extracts were demonstrated in a mouse model of acute hepatitis by reducing the levels of alanine aminotransferase (AST), alanine transaminase (ALT), alkaline phosphatase (ALP), and maldialdehyde (MDA). Additionally, black garlic extracts improved the activity of superoxide dismutase (SOD), glutathione peroxidase (GSH-Px), and glutathione reductase (GSH-Rd), while reducing tumor necrosis factor alpha (TNF-α) and interleukin-1 (IL-1β) levels in mouse liver, indicating a significant anti-inflammatory effect [55]. Increasing activity of SOD in hepatocytes during a toxic liver injury stay, in opposition to the in vitro results described above, can suggest that the influence of BG extracts on SOD activity could be dependent on the model we checked SOD activity in, that is, whether it was a simple chemical reaction or in complex model of whole organs or organisms.

The anti-inflammatory properties of black garlic were also demonstrated in a study by Lee et al. using mice with colistin-induced nephritis. An aged black garlic extract inhibited the expression of the TGF-β1 protein, which induced the NF-κB signaling pathway. CD68+ cells (*“mouse macrophages”*), infiltrated in kidneys, were also reduced, as were levels of IL-1β and TNF-α [56].

Using RT-PCR analysis, Recinella et al. examined the effect of aged black garlic extract on pro-inflammatory and pro-oxidant mediators (COX-2, TNF-α, IL-6, NF-kB) and iNOS mRNA levels on isolated LPS-stimulated heart samples. In this study, ABGE inhibited all the inflammatory and prooxidative mediators mentioned, which that suggests black garlic has anti-inflammatory properties [23]. 

Moreover, black garlic combined with vitamins D, C, and B12 had greater protective effects by inhibiting more inflammatory and oxidative pathways related to stress in LPS-exposed mice’s hearts than each vitamin alone [23]. 

The chloroform extract of aged black garlic (CEABG) inhibited TNF-α-stimulated VCAM-1 expression by decreasing reactive oxygen species (ROS) production and inhibiting the activation of the redox-sensitive transcription factor NF-κB in human umbilical vein endothelial cells (HUVEC) [57]. 

## 5. Anti-Cancer Properties of Black Garlic (BG)

Throughout the world, cancer is one of the leading causes of death. Despite scientific progress, cancer therapy remains a challenge for many types of cancer. Oncological therapies often cause unfavorable side effects for patients, leading to reductions in their quality of life. Often, it also affects the deterioration of health. 

In recent years, scientists have been focusing on researching the role of phytochemicals, mainly antioxidants, present in plants to determine new and effective methods for supporting anticancer therapy, especially in terms of limiting side effects. Intensive research is being conducted to find new substances with anti-cancer properties and therapeutic potential.

Carcinogenesis is influenced by internal and environmental factors. The consequences of the action of free radicals, e.g., reactive oxygen species generated in the human body, are one of the main internal etiological factors of cancer [36]. Oxidative stress is frequently associated with chronic inflammation, which can be followed by neighboring cell mutation and increased proliferation, often creating an environment that is conducive to the development of cancer. The antioxidative and anti-inflammatory properties of black garlic described above are also indirect anti-cancer mechanisms. In the text below, we focused on the direct anti-cancer features of aged garlic. 

It has been shown that mature black garlic extract inhibits the proliferation, migration, invasion, and metastasis of ER+ breast cancer cells in the MCF-7 and MDA-MB-361 cell lines [16]. Moreover, it stimulates apoptosis in ER+ breast cancer cells via the inhibition of the expression of anti-apoptotic proteins MCL-1 and BCL-2, while stimulating the expression of pro-apoptotic proteins BIM and BAK [16]. The reduction in MCL-1 expression was mediated by JNK activation caused by an increase in the amount of reactive oxygen species in cancer cells [16].

It has been shown that hexane extract from ripening black garlic induces apoptosis of the human leukemic cells (U937). The process of caspase-dependent apoptosis was initiated by both intrinsic and extrinsic pathways [58].

The use of ABGE in the treatment of colon cancer also has potential therapeutic value. In the study by Dong et al., it was observed that ABGE inhibited proliferation and stimulated the apoptosis of HT29 colon cancer cells. The possible mechanism of the anticancer effect is the modulation of the PI3K/Akt signaling pathway, increasing the expression of PTEN and reducing the expression of Akt and p-Akt [59].

The effects of mature black garlic on 1,2-dimethylhydrazine (DMH)-induced colon cancer models in rats and the anti-proliferative mechanisms of action were determined. Its inhibitory effect on the proliferative activity in cancerous lesions was observed without affecting the normal colonic mucosa. AGE (aged garlic extract) gradually inhibited the progression of DLD-1 by delaying cell proliferation by reducing the expression level of cyclin B1 and cdk1, which in turn was caused by the weakening of NF-κB activity [60].

The anticancer effects of aqueous extract of aged garlic on diethylnitrosamine (DEN)-induced liver cancer in rats were observed. The administration of this extract for 7 weeks resulted in a reduction in liver mass, with a significant reduction in the levels of alanine aminotransferase (ALT), aminotransferase (AST), and total bilirubin (TBIL), and an increase in antioxidant activity (TEAC test). The authors emphasize its extraordinary hepatoprotective and antioxidant effects in rats with DEN-induced liver cancer [31].

Next to S-Allyl-Cysteine (SAC), S-Allyl-Mercapto-Cysteine (SAMC) is another compound in black garlic with health-promoting properties. According to Zhang et al.’s study, SAMC induced apoptosis in human colon cancer cell line SW620 in vitro, which may explain garlic’s antiproliferative properties. Also, these results suggested that S-Allyl-Mercapto-Cysteine induces apoptosis through the JNK and p38 pathways, which activate Bax and p53 [61].

Hep-G2 cells, prostate cancer (PC-3), MCF-7 breast cancer, and mouse macrophage line (TIB-71) were inhibited by 80%–90% after 72 h by inhibiting cell proliferation and the cell cycle and causing apoptosis [35,62]. Black garlic showed dose-dependent cytotoxic effects on HL-60 leukemia cells. Unlike fresh garlic, black garlic did not induce pro-apoptotic internucleosomal DNA fragmentation; therefore, cytotoxic activity was induced in a pro-apoptotic manner only in white garlic [63]. It is worth noting that black garlic was not found to be genotoxic when tested on fruit flies (*Drosophila melanogaster*) and is even antigenotoxic [63].

Both in vitro and in vivo, black garlic extract also demonstrated anticancer and immunomodulatory properties against SGC-7901 human gastric cancer cells as well as in the mouse model. In a dose-dependent manner, ABGE significantly increased SOD and GSH-Px activity compared to the negative control. By inducing apoptosis in vitro, ABGE inhibited the growth of cancer cells [64].

On the other hand, no anti-proliferative and pro-apoptotic effects were demonstrated in the MCF-10A breast cancer cell line [16]. In a study on three lung cancer cell lines (H1975, H520, A549), water and alcohol extracts also showed negligible anti-proliferative properties [46]. A study by Kim et al. found that black garlic extract had no cytotoxic effects when applied to RAW264.7 and RBL-2H3 cells [8]. 

The effects of black garlic on anti-cancer activity and their potential molecular mechanism are summarized in Table 3.

Angiogenesis is an important process in tumor development and the inhibition of new blood vessel formation could be effective in cancer treatment. Arianingrum R et al. performed in vivo experiments using the Chorio-Allantoic Membrane Assay (CAM) for the evaluation of blood vessel formation as well as in silico study via docking analysis of BG compounds on the VEGF receptor [65]. The authors showed that ethanol extract, ethyl acetate, and n-hexane fractions of BG inhibit angiogenesis, with n-hexane fraction having the highest efficacy. The molecular docking assay indicated some possible inhibitory interaction between the black garlic bioactive compounds and VEGFR. On the other hand, the induction of angiogenesis was described in vivo in a zebrafish model or male Wistar rat model and in vitro with the use of HUVEC cells [66,67].

The described above basic and preclinical studies showed that ABGE could directly influence ER+ breast cancer cells, some leukemic cells, lung cancers, colon cancers, liver cancers, and prostate cancers. On the other hand, some breast and lung cancer cell lines were resistant to and did not undergo apoptosis when treated with BG extract. Considering breast and lung cancer, the results are contradictory, and so further research will be focused on the type of breast and lung cancer, as well as their molecular characteristics associated with resistance to ABGE. The new experiments considering tumor cells resistant to approved anti-cancer agents shall be conducted and answer whether BG extracts can induce apoptosis in these cells or not.

## 6. Conclusions

The phytochemicals isolated from the different plants are tested in pre-clinical studies or have been already used in treatment e.g., paclitaxel isolated from Pacific yew. We suggested that, with a wide range of valuable chemical compounds, black garlic may be an interesting source of active agents leading to drug development.

Black garlic cloves contain health-promoting ingredients with antioxidant and anti-inflammatory properties. The significant effect of black garlic on the treatment of acute and chronic inflammation has been shown.

The studies have shown that black garlic compounds can indirectly (through reduction in oxidative stress and chronic inflammation) and directly inhibit development and progression of malignant tumors, for instance.

Considering the influence of black garlic extracts on cancer cells, metabolic pathways, and growth mechanisms, along with the absence of pro-mutagenic effects, black garlic extracts could support modern oncological treatment as adjuvant therapy with few to no side effects. As the anticancer effect of BG extracts is probably strongly associated with antioxidant and anti-inflammatory mechanisms, we suggest the use of ABGE in mainly the prevention of cancer development and the supportive treatment of chronic inflammation. Because of that and based on the results of preclinical studies, aged garlic extracts shall be used for the development and manufacturing of pharmaceuticals designed for oral delivery. The future studies considering particular purified chemical compounds of BG and their direct influence on cancer cells could be the basis for the development of medicaments for primary cancer treatment, if efficacy in doses suitable for human use were confirmed. 

## Figures and Tables

**Figure 1 ijms-25-01801-f001:**
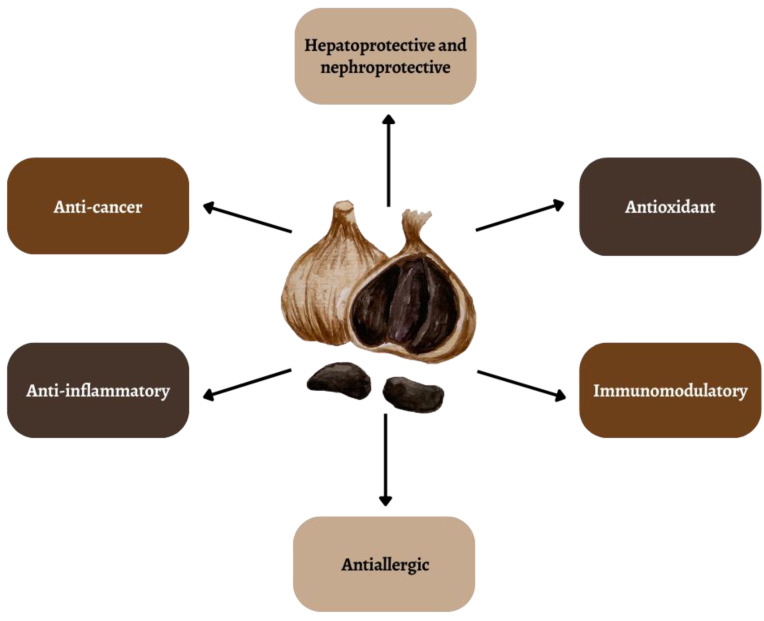
Selected properties of black garlic.

**Figure 2 ijms-25-01801-f002:**
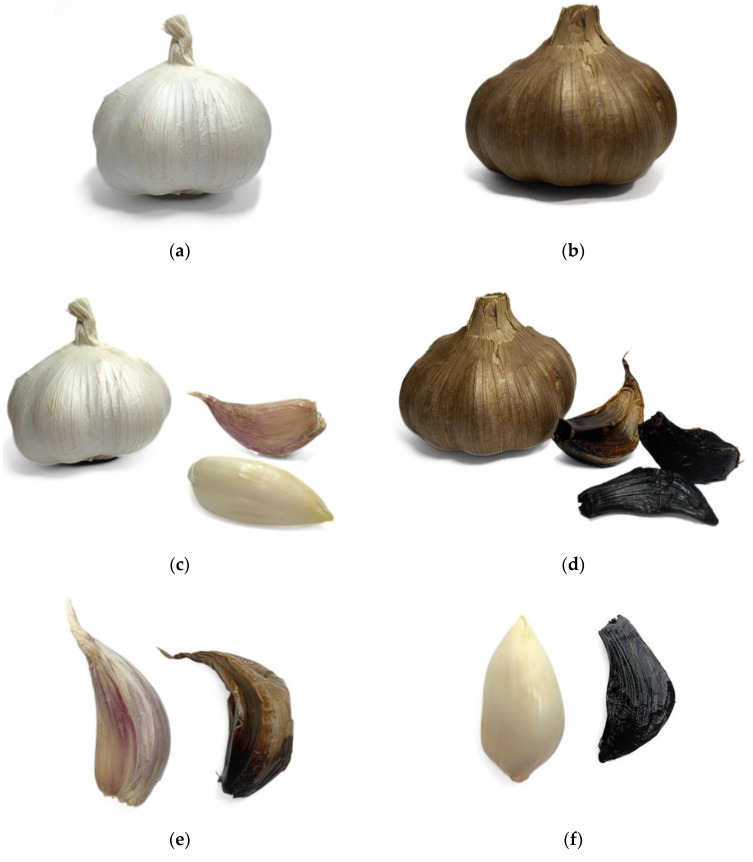
Fresh garlic and black garlic heads and cloves comparison: (**a**) fresh garlic (*Allium sativum*) head; (**b**) black garlic (*Allium sativum*) head; (**c**) fresh garlic head and gloves; (**d**) black garlic head and cloves; (**e**,**f**) a side-by-side comparison of a clove of fresh garlic (left) and black garlic (right) (photograph by Julia Trojniak).

**Figure 3 ijms-25-01801-f003:**
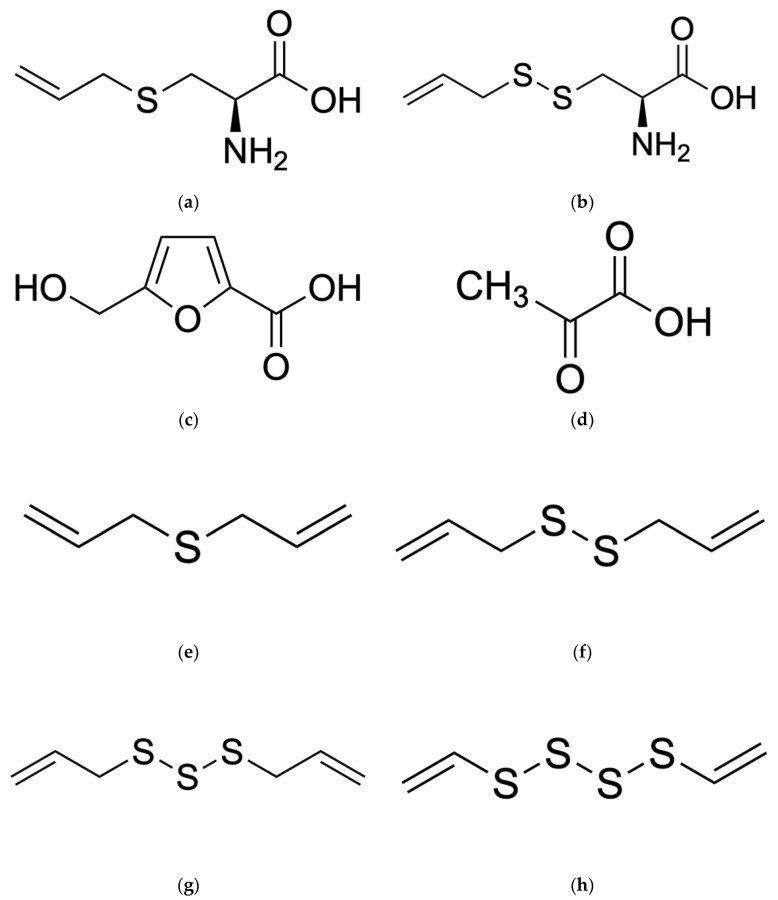
Chemical structures of several of the most important compounds found in black garlic (BG): (**a**) S-Allyl-Cysteine (SAC); (**b**) S-Allyl-Mercapto-Cysteine (SAMC); (**c**) 5-Hydroxymethylfurfural (5-HMF); (**d**) pyruvate/pyruvic acid; (**e**) dialyl sulfides (DAS); (**f**) dialyl disulfides (DADS); (**g**) dialyl trisulfides (DATS); (**h**) dialyl tetrasulfide.

**Table 1 ijms-25-01801-t001:** Fresh garlic (FG) and black garlic (BG) chemical composition comparison.

Component		Fresh Garlic (FG)	Black Garlic (BG)	Change in Substance Content in Black Garlic Compared to Fresh Garlic
Moisture (%)		62.00 ± 7.00; 64.70 ± 0.50; 74.00 ± 2.00 (g/100 g FW ^1^) [25]	54.00 ± 3.00 (g/100 g FW) [25]	↓
		66.64 ± 1.31 [26]	58.20 ± 0.39 [26]	
		62.31 ± 0.57 [27]	39.03 ± 0.01–58.48 ± 1.18 [27]	
		60.30 [28]	45.10 [28]	
		62.86 ± 0.13 [29]	43.03 ± 0.42 [29]	
		59.57 ± 2.44 [13]	38.26 ± 4.48 [13]	
Dry matter (%)		32.13 ± 0.99 [30]	52.41 ± 0.92 [30]	↑
		37.14 ± 0.13 [29]	56.96 ± 0.42 [29]	
pH		6.20 ± 0.03 [30]	3.95 ± 0.03 [30]	↓
		6.37 ± 0.04 [29]	3.94 ± 0.05 [29]	
		6.04 ± 0.04 [13]	4.39 ± 0.61 [13]	
Nutritional characteristics	Fats	0.47 ± 0.02; 0.67 ± 0.03; 0.74 ± 0.02 (g/100 g FW) [25]	0.722 ± 0.001 (g/100 g FW) [25]	↑
		0.18 ± 0.01 (%) [26]	0.58 ± 0.11 (%) [26]	
		0.1 (%) [28]	0.3 (%) [28]	
	Amino acids	843.11 ± 3.75 (mg/100 g FW) [27]	167.65 ± 1.08–372.88 ± 2.23 (mg/100 g FW) [27]	↓
	Protein	6.50 ± 0.10; 7.80 ± 0.20; 5.20 ± 0.10 (g/100 g FW) [25]	7.40 ± 0.10 (g/100 g FW) [25]	↑
		0.70 ± 0.02 (%) [26]	0.97 ± 0.07 (%) [26]	
		8.40 (%) [28]	9.10 (%) [28]	
	Carbohydrates	28.00 ± 2.00; 24.20 ± 0.40; 18.00 ± 2.00 (g/100 g FW) [25]	35.00 ± 3.00 (g/100 g FW) [25]	↑
		28.7% [28]	47% [28]	
	Ash	2.90 ± 0.10; 2.70 ± 0.20; 1.60 ± 0.04 (g/100 g FW) [25]	3.20 ± 0.10 (g/100 g FW) [25]	↑
		0.92 ± 0.62 (%) [26]	1.81 ± 0.05 (%) [26]	
		73.59 ± 0.89 (mg/100 g) [27]	75.36 ± 0.02–114.36 ± 8.65 (mg/100 g) [27]	
		ND [28]	2.10 (%) [28]	
	Energy (kcal/100 g FW ^1^)	141.00 ± 8.00; 134.00 ± 2.00; 100.00 ± 9.00 [25]	177.00 ± 8.00 [25]	↑
		138.00 [28]	227.10 [28]	
Free sugars	Total	1.32 ± 0.05; 1.48 ± 0.01; 0.70 ± 0.01 (g/100 g FW) [25]	33.6 ± 0.7 (g/100 g FW) [25]	↑
		4.47 ± 0.11 (%) [26]	6.19 ± 0.02 (%) [26]	
		292.54 ± 2.01 (mg/100 g) [27]	754.51 ± 4.05–4726.04 ± 15.74 [27]	
	Xylose	0.82 ± 0.01 (g/100 g FW) [25]	ND ^2^	↓
	Fructose	0.45 ± 0.01; 0.09 ± 0.01; 0.20 ± 0.01 (g/100 g FW) [25]	30.4 ± 0.7 (g/100 g FW) [25]	↑
		63.89 ± 3.42 (mg/100 g) [26]	2043.73 ± 4.99 (mg/100 g) [26]	
		7.07 ± 0.08 (mg/g) [22]	40.02 ± 0.71 (mg/g) [22]	
		31.40 ± 0.96 (mg/100 g) [27]	486.75 ± 11.72–3383.23 ± 44.03 (mg/100 g) [27]	
		9.36 ± 0.13 (g/100 g DW ^3^) [30]	31.05 ± 1.34 (g/100 g DW) [30]	
	Glucose	0.28 ± 0.02; 0.04 ± 0.01; 0.12 ± 0.01 (g/100 g FW) [25]	2.14 ± 0.03 (g/100 g FW) [25]	↑
		2.12 ± 0.05 (g/100 g DW) [26]	4.84 ± 0.28 (g/100 g DW) [26]	
	Saccharose	0.58 ± 0.01; 1.35 ± 0.01; 0.38 ± 0.01 (g/100 g FW) [25]	0.23 ± 0.05 (g/100 g FW) [25]	↓/↑
		76.31 ± 0.05 (mg/100 g) [26]	119.14 ± 3.51 (mg/100 g) [26]	
		0.02 ± 0.00 (g/100 g DW) [30]		
Total Phenolic Content (TPC)		0.59 ± 0.08 (mg/100 g) [26]	1.56 ± 0.14 (mg/100 g) [26]	↑
		3.65 ± 0.17 (mg GAE ^4^ 100 g^−1^) [8]	22.17 ± 0.75 (mg GAE 100 g^−1^) [8]	
		3.82 ± 0.37 [13]	14.03 ± 4.44 [13]	
Total Flavonoid Content (TFC)		0.14 ± 0.01 (mg/100 g) [26]	0.77 ± 0.03 (mg/100 g) [26]	↑
		37.75 ± 0.54 (mg/100 g DW) [30]	57.80 ± 0.55 (mg/100 g DW) [30]	
Pyruvate (μM/g)		486.71 ± 12.08 [22]	2456.54 ± 23.93 [22]	↑
		188.47 ± 3.03 [26]	277.85 ± 2.57 [26]	
S-Allyl-Cysteine (SAC)		1.24 ± 9.22 (mg/g) [22]	2.12 ± 10.17 (mg/g) [22]	↑
		42.7 (μg/g) [31]	656.5 (μg/g) [31]	
		23.7 (μg/g) [28]	194.3 (μg/g) [28]	
5-Hydroxymethylfurfural (5-HMF)		ND	4.82 ± 0.06 (g/kg) [32]	↑
			0.25 ± 0.04 (g/kg) FM ^5^ [33]	
			6–8 (g/kg) [24]	
Thiosulfan		6.50 ± 0.29 (μM/g) [22]	91.22 ± 0.54 (μM/g) [22]	↑
Allicin		3.62 ± 0.01 (mg/g) [22]	ND	↓
Vitamins		6632.91 ± 18.62 mg/kg [34]	7618.24 ± 28.47–9010.44 ± 30.61 mg/kg [34]	↑
Minerals (mg/100 g)		567.88 ± 4.48 [26]	969.12 ± 19.31 [26]	↑
		1173.50 ± 2.43 [27]	1314.68 ± 2.76–1337.71 ± 2.77 [27]	

^1^ FW—fresh weight; ^2^ ND—not determined; ^3^ DW—dry weight; ^4^ GAE—gallic acid equivalents; ^5^ FM—fresh matter, ↑-increase, ↓-decrease.

**Table 2 ijms-25-01801-t002:** An analysis of the change in composition and ability of selected substances affecting black garlic antioxidant properties during aging.

Method	Aging Conditions		Aging Period (Weeks)					Source
			0	1	2	3	4	
	Temperature	Moisture						
Total Phenolic Content (mg GAE ^1^/g)	70 °C	90%	13.91 ± 1.62	25.81 ± 1.59	35.28 ± 0.32	58.33 ± 1.90	55.25 ± 0.70	[39]
			4.62 ± 0.48	-	11.84 ± 0.14	23.43 ± 0.41	27.08 ± 0.14	[40]
	70 °C	85%	5.85 ± 0.14	5.98 ± 0.16	7.45 ± 0.22	9.89 ± 0.21	15.48 ± 0.53	[41]
	72 ± 2 °C	∼90%	3.26 ± 0.29; 4.86 ± 0.24; 3.56 ± 0.2	-	4.42 ± 0.2; 6.75 ± 0.3; 9.97 ± 0.91	10 ± 0.4; 13.64 ± 0.52; 9.36 ± 0.25	12.63 ± 0.26; 15.79 ± 0.41; 12.65 ± 0.64	[42]
Total Flavonoid Content (mg RE ^2^/g)	70 °C	90%	3.22 ± 0.07	5.38 ± 0.06	8.34 ± 0.61	15.37 ± 0.52	16.26 ± 1.69	[39]
			0.86 ± 0.03	-	2.48 ± 0.05	7.27 ± 0.10	8.75 ± 0.21	[40]
S-Allyl-Cysteine (SAC) (mg/100 g DW ^3^)	60 °C	65%	104.34 ± 10.38	1772.15 ± 48.98	313.22 ± 63.75	387.42 ± 17.25	174.65 ± 9.65	[18]
	60 °C	80%		1750.29 ± 49.63	360.00 ± 33.43	228.79 ± 31.94	200.54 ± 6.19	
	80 °C	65%		654.50 ± 22.95	104.24 ± 14.77	113.43 ± 2.197	ND	
	80 °C	80%		874.26 ± 57.27	123.97 ± 14.19	41.53 ± 4.49	4.34 ± 0.09	
(5-HMF) (mg/100 g DW)	60 °C	65%	ND ^4^	ND	ND	1.81 ± 0.05	2.34 ± 0.02	[18]
	60 °C	80%				1.19 ± 0.06	3.20 ± 0.03	
	80 °C	65%		57.44 ± 0.29	673.41 ± 7.62	832.13 ± 1.46	511.24 ± 0.88	
	80 °C	80%		50.03 ± 0.04	724.60 ± 0.95	1721.41 ± 4.17	400.10 ± 0.23	
DPPH test (%)	60 °C	65%	146.96 ± 36.09	146.29 ± 25.72	374.55 ± 25.80	944.95 ± 35.92	1153.14 ± 44.61	[18]
	60 °C	80%		150.11 ± 15.98	413.07 ± 23.97	953.70 ± 36.60	1290.14 ± 61.62	
	80 °C	65%		4308.06 ± 114.87	4814.81 ± 127.55	4163.51 ± 205.50	3275.61 ± 154.13	
	80 °C	80%		5390.02 ± 180.03	5643.58 ± 61.98	5194.55 ± 197.98	3969.23 ± 275.51	
	70 °C	90%	20.27 ± 0.13	-	50.49 ± 0.47	82.49 ± 0.26	90.98 ± 0.23	[40]
ABTS test (%)	60 °C	65%	2768.94 ± 176.53	4681.05 ± 338.07	4862.37 ± 15.73	6564.28 ± 91.96	8753.91 ± 200.25	[18]
	60 °C	80%		5836.87 ± 151.82	4972.23 ± 271.51	7350.95 ± 151.56	8534.50 ± 433.20	
	80 °C	65%		20,405.95 ± 858.88	22,759.16 ± 912.09	20,614.96 ± 598.57	16,611.14 ± 312.57	
	80 °C	80%		25,421.11 ± 262.39	22,112.16 ± 856.43	23,300.72 ± 278.24	20,644.76 ± 776.78	
	72 ± 2 °C	∼90%	23.9 ± 2.79; 62.61 ± 3.37; 32.62 ± 2.76	-	52 ± 2.5; 81.85 ± 3.52; 116.14 ± 4.49	103.11 ± 4.0; 135.74 ± 5.37; 114.31 ± 4.98	97.21 ± 1.6; 113.77 ± 5.49; 81.99 ± 3.32	[42]

^1^ GAE—gallic acid equivalents; ^2^ RE—rutin equivalents; ^3^ DW—dry weight; ^4^ ND—not determined.

**Table 3 ijms-25-01801-t003:** The effects of black garlic in anti-cancer activity and its potential mechanism.

Type of Malignancy	Cell Line	In Vitro/Animal Model	Effect	Possible Molecular Mechanism	Reference
ER+ breast cancer	MCF-7 and MDA-MB-361	in vitro	inhibition of the proliferation, migration, invasion and metastasis, induction of apoptosis	inhibition of the expression of anti-apoptotic proteins MCL-1 and BCL-2, while stimulating the expression of pro-apoptotic proteins BIM and BAK	[16]
histiocytic lymphoma	U937	in vitro	concentration- and time-dependent growth inhibition by induction of apoptosis,	upregulation of death receptor 4 and Fas legend, and an increase in the ratio of Bax/Bcl-2 protein expression	[58]
colon cancer	HT29	in vitro	induction of apoptosis and cell cycle arrest	regulation of the function of the PI3K/Akt pathway through upregulating PTEN and downregulating Akt and p-Akt expression, as well as suppressing its downstream target, 70-kDa ribosomal protein S6 kinase 1, at the mRNA and protein levels	[59]
colorectal cancer	1,2-dimethylhydrazine (DMH)-induced primary cancer, DLD-1 and MRC-5 cell lines	animal model (F344 rats) and in vitro study	decrease in the number of aberrant crypt foci, suppression of the proliferative activity in adenoma and adenocarcinoma lesions, but no effect on normal colon mucosa.	delayed cell cycle progression by downregulating cyclin B1 and cdk1 expression via inactivation of NF-κB in the human colorectal cancer cells but no induction of apoptosis.	[60]
colorectal cancer	SW620	in vitro	reduction in the malignant cell viability in a dose- and time-dependent manner, partially through the induction of apoptosis	induction of apoptosis through JNK and p38 signaling pathways, increased tumor protein p53 (p53), and Bax activation	[61]
breast cancer, prostate cancer, liver cancer, and colon cancer	MCF-7 (breast) PC-3 (prostate) Hep-G2 (liver) Caco-2 (colon)	in vitro	inhibition of cell proliferation was observed for Hep-G2, MCF-7, TIB-71, and PC-3 cells, induction of apoptosis, and G1 cell cycle arrest.	activation of caspase-3 and -7	[62]
acute promyelocytic leukemia	HLA-50	in vitro	strong cytotoxic effect and induction of DNA proapoptotic internucleosomal fragmentation against	not examined	[63]
gastric cancer	SGC-7901	in vitro and animal model using male white mice of the Kunming strain inoculated with murine forestomach cells	dose-dependent induction of apoptosis growth inhibition of inoculated tumors	significantly increased the activity of SOD and GSH-Px in a dose-dependent manner, increased activity of Il-2	[64]

## Data Availability

No new data were created.

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
