# Peer review of "Anti-Cancer and Anti-Inflammatory Properties of Black Garlic"

_ijms, 2024, doi:10.3390/ijms25031801_

Round 1
Reviewer 1 Report
Comments and Suggestions for Authors
This manuscript reviewed the relevant research progress on the anti-inflammatory and anticancer effects of black garlic. Overall it is in good writing. It is suggested that the following should be revised.
1. A section entitled "Active compounds of black garlic" should be added after the introduction section, which mainly introduces the composition and content of active compounds contained in black garlic.
2. The logic of the anti-cancer section is not strong, just a simple list of other research results. It is recommended to add a paragraph to state what cancers black garlic can resist, and then write the corresponding research progress of each cancer. It is recommended to add a table entitled “The effects of black garlic in anti-cancer activity and its potential mechanism” in the anti-cancer section. The contents of the table include cancer types, models, active ingredients of black garlic and anti-cancer mechanism.
3. The conclusion is too tedious. It is suggested to simplify the content. The conclusion section should be coded as "4. Conclusion".
4. Line 9, format alignment.
5. Table 1. “62 ± 7; 64,7 ± 0,5; 74 ± 2 (g/100 g FW1) ……”, accurate the digits to 2 decimals; change 64,7 to 64.7; the same suggestion with others. What is the unit of amino acids 843,11 ± 3,75?
6. Line 92, delete “:”.
7. Is there a contradiction between line 112-114 and 174? Explain.
8. Line 283, in-volving, delete “-”.
9. Unify the format of references according to journal’s requirements. Such as references 1.
10. Line 260, “As a wide range of valuable chemical compounds”. Change as to with.
Comments on the Quality of English LanguageThe quality of the manuscript is qutie good in English writing. Only minor revisation is required.
Author Response
Dear Sirs,
We would like to thank the Reviewers for comprehensive and helpful evaluation of our manuscript. The Reviewers concerns about our paper were very helpful and we have revised the paper according to their suggestion. We have implemented all comments of the Reviewers and hereby send you the new version of manuscript as well as a letter including all answers and text changes. We also provide revised manuscript file in Track Changes' mode.
On behalf of the authors,
Jacek Tabarkiewicz
Reviewer 1
This manuscript reviewed the relevant research progress on the anti-inflammatory and anticancer effects of black garlic. Overall it is in good writing. It is suggested that the following should be revised.
We would like to thank Reviewer for all valuable comments and suggestion.
- A section entitled "Active compounds of black garlic" should be added after the introduction section, which mainly introduces the composition and content of active compounds contained in black garlic.
We appreciate this comment. Section entitled "Active compounds of black garlic" is added.
- The logic of the anti-cancer section is not strong, just a simple list of other research results. It is recommended to add a paragraph to state what cancers black garlic can resist, and then write the corresponding research progress of each cancer. It is recommended to add a table entitled “The effects of black garlic in anti-cancer activity and its potential mechanism” in the anti-cancer section. The contents of the table include cancer types, models, active ingredients of black garlic and anti-cancer mechanism.
This is valuable comments. The cited below part of text was added into chapter describing anti-cancer properties of BG “The described above basic and preclinical studies has shown that ABGE could directly influence ER+ breast cancer cells, some leukemic cells, lung cancers, colon cancers, liver cancers, and prostate cancers. On the other hand, some breast and lung cancer cell lines were resistant to and did not undergo apoptosis when treated with BG extract. Considering breast and lung cancer, the results are contradictory, so further research shall be focused on the type of breast and lung cancer as well as their molecular characteristics associated with resistance to ABGE. The new experiments considering tumor cells resistant to approved anti-cancer agents shall be conducted and answer if BG extracts can induce apoptosis in these cells or not.”
The suggested table is added as Table 3
- The conclusion is too tedious. It is suggested to simplify the content. The conclusion section should be coded as "4. Conclusion".
We thank to Reviewer for this comments. Number of Conclusions paragraph was corrected to 6, due to added sections according to Reviewers’ suggestions.
- Line 9, format alignment.
Corrected.
- Table 1. “62 ± 7; 64,7 ± 0,5; 74 ± 2 (g/100 g FW1) ……”, accurate the digits to 2 decimals; change 64,7 to 64.7; the same suggestion with others. What is the unit of amino acids 843,11 ± 3,75?
We would like to thank Reviewer for this suggestion. All values are accurate to 2 decimals. All “,” were corrected to “.”. Unit was added for amino acids content.
- Line 92, delete “:”.
We thank Reviewer for spotting this editorial mistake. Corrected.
- Is there a contradiction between line 112-114 and 174? Explain.
Results described in lines 112-114 are based on in vitro study based on chemical experiments, on the other hand study described in line 174 is complex study with the use of mouse model of toxic liver injury. We describe this difference as well as make comment about contradictory results from different types of experiments.
- Line 283, in-volving, delete “-”.
We thank Reviewer for spotting this editorial mistake. Corrected.
- Unify the format of references according to journal’s requirements. Such as references 1.
References are unified and prepared according to https://mdpi-res.com/data/mdpi_references_guide_v5.pdf. As well as Example: 15. Díaz, D.D.; Converso, A.; Sharpless, K.B.; Finn, M.G. 2,6-Dichloro-9-thiabicyclo[3.3.1]nonane: Multigram Display of Azide and Cyanide Components on a Versatile Scaffold. Molecules 2006, 11, 212–218, doi:10.3390/11040212.
- Line 260, “As a wide range of valuable chemical compounds”. Change as to with.
Corrected according to suggestion.

Reviewer 2 Report
Comments and Suggestions for Authors
Comments#
The authors described scientific evidence of the black garlic(BG) such as scavenge free radical, anti-inflammation and anti-cancer effect. The authors also described several phytochemicals may be important for the effects. This review paper provides scientific interest for wider field such as nutrition, immunity, cancer and research. However, there are some points to need to reconsider for publication.
Table1
This table is not well aligned, so it is difficult to recognize in detail. Authors should be modified this table to be published.
1. Introduction
L.70
“the technological parameters of aging” is difficult to understand for readers. Please consider adding additional explanation.
2. Anti-inflammatory properties of black garlic (BG)
In this section, the authors described two topics (i.,e, anti-oxidation and anti-inflammation properties of BG), in addition this part is too long. So, the authors should describe on two topics, separately.
3. Anti-cancer properties of black garlic (BG)
Authors described the phytochemicals have antiangiogenic properties, however there are no description in text. Authors describe in detail for the readers who are working on cancer research field.
L.198
“significantly” is not best word. I recommend “.., leading to reduce their quality of life”
L.200-204
Some phytochemicals were found and used for clinical and pre-clinical studies, such as paclitaxel. Authors described a phytochemical(s) for anti-cancer drug(s) with reference.
L.205
“both internal and external factors contribute to this complex process.” This phase is ambiguity. Authors reconsider this part.
L.224
“colon cancer in rats” will be “colon cancer model in rats”
L.225
“mechanism of proliferation” may be “mechanism of anti-proliferation”
L.227
AGE needs spell out.
L.250
“In vitro” may be “In vivo”, please check it out.
Author Response
Dear Sirs,
We would like to thank the Reviewers for comprehensive and helpful evaluation of our manuscript. The Reviewers concerns about our paper were very helpful and we have revised the paper according to their suggestion. We have implemented all comments of the Reviewers and hereby send you the new version of manuscript as well as a letter including all answers and text changes. We also provide revised manuscript file in Track Changes' mode.
On behalf of the authors,
Jacek Tabarkiewicz
Reviewer 2
The authors described scientific evidence of the black garlic(BG) such as scavenge free radical, anti-inflammation and anti-cancer effect. The authors also described several phytochemicals may be important for the effects. This review paper provides scientific interest for wider field such as nutrition, immunity, cancer and research. However, there are some points to need to reconsider for publication.
We would like to thank Reviewer for reviewing our manuscript.
Table1
This table is not well aligned, so it is difficult to recognize in detail. Authors should be modified this table to be published.
The table is modified.
- Introduction
L.70
“the technological parameters of aging” is difficult to understand for readers. Please consider adding additional explanation.
Corrected.
- Anti-inflammatory properties of black garlic (BG)
In this section, the authors described two topics (i.,e, anti-oxidation and anti-inflammation properties of BG), in addition this part is too long. So, the authors should describe on two topics, separately.
We describe , these topics together, because a strong link between oxidative stress and inflammation, but now are divided according to suggestion.
- Anti-cancer properties of black garlic (BG)
Authors described the phytochemicals have antiangiogenic properties, however there are no description in text. Authors describe in detail for the readers who are working on cancer research field.
We appreciate this comment The paragraph describing influence of BG compounds on angiogenesis as added.
L.198
“significantly” is not best word. I recommend “.., leading to reduce their quality of life”
Corrected.
L.200-204
Some phytochemicals were found and used for clinical and pre-clinical studies, such as paclitaxel. Authors described a phytochemical(s) for anti-cancer drug(s) with reference.
Paclitaxel was isolated from Pacific yew not from the black garlic. We agreed that some phytochemicals are used for clinical and pre-clinical studies, on the other hand this review is focused on the black garlic and describing phytochemicals isolated form other plants is not directly associated with topic of this review. Describing phytochemicals isolated form other plants in paper entitled “Anti-cancer and anti-inflammatory properties of black garlic” could be confounding for readers. We mentioned in Conclusions section that phytochemicals are promising agents in treatment of human diseases.
L.205
“both internal and external factors contribute to this complex process.” This phase is ambiguity. Authors reconsider this part.
We appreciated this comment. We rewrite these sentences in more unambiguous way.
L.224
“colon cancer in rats” will be “colon cancer model in rats”
We thank Reviewer for this comment. Corrected.
L.225
“mechanism of proliferation” may be “mechanism of anti-proliferation”
Corrected.
L.227
AGE needs spell out.
Corrected.
L.250
“In vitro” may be “In vivo”, please check it out.
We really appreciated this comment. Checked out and corrected.

Reviewer 3 Report
Comments and Suggestions for Authors
The current review article is an interesting manuscript that focuses on anticancer and anti-inflammatory properties of black garlic. It appears to be well written and robust within the topic, hence I only advise for the following alterations before acceptance for publication:
- A bridge should be made between anticancer and anti-inflammatory properties, since inflammation has been shown to have a strong link to cancer pathogenesis, even being considered as a potential therapeutic target against cancer, hence this should be discussed;
- In each analyzed study, more should be said about it, for example in “Studies conducted on the LPS-adapted macrophage line (RAW264.7) showed that pyruvate present in black garlic inhibits the formation of ROS induced by H2O2 [10].”, the used methodology and respective results should be further mentioned, namely the actual numbers, controls, assays’ characteristics, etc.;
- Given the apparent great potential of the mentioned extracts and included compounds, the authors should mention what type of cancers this should be best suited for, what kind of administration routes, what type of formulations, and whether the authors consider these results as good enough so that black garlic derived compounds could be used in the future as primary cancer therapies, or just as adjuvant therapies;
- In the introduction section, an image should added including a schematization and summary of black garlic therapeutical potential;
- An abbreviation list is missing and should be added.
Author Response
Dear Sirs,
We would like to thank the Reviewers for comprehensive and helpful evaluation of our manuscript. The Reviewers concerns about our paper were very helpful and we have revised the paper according to their suggestion. We have implemented all comments of the Reviewers and hereby send you the new version of manuscript as well as a letter including all answers and text changes. We also provide revised manuscript file in Track Changes' mode.
On behalf of the authors,
Jacek Tabarkiewicz
Reviewer 3
The current review article is an interesting manuscript that focuses on anticancer and anti-inflammatory properties of black garlic. It appears to be well written and robust within the topic, hence I only advise for the following alterations before acceptance for publication:
We would like to thank Reviewer for all valuable comments and suggestion.
- A bridge should be made between anticancer and anti-inflammatory properties, since inflammation has been shown to have a strong link to cancer pathogenesis, even being considered as a potential therapeutic target against cancer, hence this should be discussed;
We appreciate this valuable comment. We added part of text as below: “The oxidative stress is frequently associated with chronic inflammation, which can be followed by neighboring cell mutation and increased proliferation, often creating an environment that is conducive to the development of cancer. The described above anti-oxidative and anti-inflammatory properties of black garlic are also indirect anti-cancer mechanisms. In the text below we focused on direct anti-cancer features of the aged garlic.”
- In each analyzed study, more should be said about it, for example in “Studies conducted on the LPS-adapted macrophage line (RAW264.7) showed that pyruvate present in black garlic inhibits the formation of ROS induced by H2O2 [10].”, the used methodology and respective results should be further mentioned, namely the actual numbers, controls, assays’ characteristics, etc.;
Several data from the studies were included according to suggestion.
- Given the apparent great potential of the mentioned extracts and included compounds, the authors should mention what type of cancers this should be best suited for, what kind of administration routes, what type of formulations, and whether the authors consider these results as good enough so that black garlic derived compounds could be used in the future as primary cancer therapies, or just as adjuvant therapies;
This valuable comment. We discussed this issue in Conclusions.
- In the introduction section, an image should added including a schematization and summary of black garlic therapeutical potential;
We would like to thank for this comment. We moved figure 3 to introduction section, which summarizes the black garlic therapeutical potential.
- An abbreviation list is missing and should be added.
Abbreviations list is added.
